# Inhibitory Risks Affecting the Maintenance of Healthy Lifestyle Habits—A Study Based on Demographic Factors and Personality Traits

**DOI:** 10.3390/ijerph16132322

**Published:** 2019-06-30

**Authors:** Wenzhu Cui, Akane Katayama, Hiroki Shimizu, Mamoru Taniguchi

**Affiliations:** 1Laboratory for Urban Transformation, Systems and Information Engineering, University of Tsukuba, Tsukuba 305-8573, Ibaraki, Japan; 2Laboratory for Urban Transformation, College of Policy and Planning Sciences, University of Tsukuba, Tsukuba 305-8573, Ibaraki, Japan; 3Department of Policy and Planning Sciences, University of Tsukuba, Tsukuba 305-8573, Ibaraki, Japan

**Keywords:** stages of behavioral change, living habits, living environment, healthy lifestyle, personality traits, demographic factors

## Abstract

People with different personal attributes, both intrinsic (personality traits) and extrinsic (demographic factors), perceive different inhibitions in the maintenance of a healthy lifestyle. This study examines the correlation between inhabitants’ personal attributes, internal and external, and their perception of risk factors in the maintenance of a healthy lifestyle. Results show that (1) residents prone to ‘challenge’ and ‘commitment’ are more sensitive to the inhibitory risks. (2) Inhabitants under 65 years old with ‘challenge’ in their personality are more likely to report ‘time’ as a constraining factor. (3) Regarding walking habits, residents are not only impacted by the living environment but also restricted by the demographic factors of economics and time.

## 1. Introduction

### 1.1. Background

Recently, with the declining birth rate and rapid aging, the emergence of social problems such as labor shortages and increased medical expenses incurred by the government have become urgent issues for residents in economically developed countries [1]. Also, health problems such as the collapse of nutritional balance and lack of exercise have become more severe with changes in the living environment. This includes the expansion of food deserts, making fast-food and other unhealthy options more convenient, and inadequate use of public transportation due to increased automobile use. Therefore, the risks of lifestyle diseases such as obesity and diabetes are increasing worldwide [2].

To address these diseases, health science is generally used to intervene in treating lifestyle diseases and to manage the daily habits of residents [3]. These methods can be targeted at individual residents, but solving such problems on a population scale is arduous. Additionally, preventing the recurrence of lifestyle diseases is challenging to accomplish merely by relying solely on personal efforts. Therefore, it has been deduced that a healthy city and policies supporting an improved living environment will aid residents in maintaining healthy living habits [4].

Efforts must be taken to prevent the deterioration of healthy living habits at an early stage, without merely waiting for the occurrence of lifestyle diseases [5]. In Japan, the policy of ‘Kenkou-Nippon 21 (Second)’ was put forward in 2011, emphasizing the need for measures that might improve living habits [6]. Among these policy propositions was the promotion of six living habits: ‘reduction in the drinking of alcohol’, ‘breakfast’, ‘walking’, ‘nutritional balance’, ‘exercise’ and ‘sleep’, of which the latter four are addressed in this paper. Specific improvement targets have been established for living habits, such as ‘increasing the number of residents with a quality diet’ and ‘increasing the number of steps walked daily’. Moreover, publications of the policy development status, the progress of each target, and assessments of the degree of achievement have been done [7]. That being said, although various improvement policies are being implemented, according to the latest ‘Kenkou-Nippon 21’ Interim Report, the achievement of specific targets has remained low or is deteriorating [8]. It appears that some living habits cannot be easily changed. Although residents might attempt to improve these living habits, they give up, and eventually, a ‘reversion’ or ‘rebound’ phenomenon occurs [9].

In the field of public health, to support the improvement of living habits such as smoking cessation and exercise, behavioral changes are divided into five stages, from ‘pre-contemplation’ to ‘contemplation’, ‘preparation’, and ‘action’, up until the ‘maintenance’ period [10]. Several studies have demonstrated that living habit improvement is achieved better through the introduction of corresponding measures aimed at each particular stage of behavioral change [11,12]. Based on this concept, research specifically addressing improvement policies has been extended into other living habits such as nutritional balance [13]. Existing research elucidates to the relationship between the demographic factors of demographics, such as age and employment, and stages of behavioral change. Simultaneously, other studies explore whether differences in personality traits, such as self-efficacy and the Big-Five personality traits ([14] (5 traits: ‘openness’, ‘conscientiousness’, ’extraversion’, ‘agreeableness’, ‘neuroticism’)), affect the stages of behavioral change [15,16]. Thus, to formulating effective policies that support the sustainability of healthy living habits in the field of urban planning, emphasizing concepts of the stage of behavioral change is essential. Furthermore, plans that are based on the research of individual residents must be transformed to produce policies that are more feasible, efficient, and can be applied to large-scale population regions as well.

In the field of urban planning, particularly addressing walking habits, numerous studies have examined and applied the concept of the stages of behavioral change, such as that of health and mobility management in Japan [17,18]. Various studies have analyzed residents in the ‘pre-contemplation’ or ‘prepared’ stages. Nevertheless, few reports of the literature describe the ‘reversion’ stage that is likely to occur. Some studies of other living habits specifically examine the current situation of stages [19]. Earlier studies applied various analyses for different demographic factors such as age and gender. However, studies examining personality traits are emphasized in the field of public health. Regarding the inhibitory risk of maintaining living habits, even if demographic factors are the same, the ability to suppress inhibitory risks varies according to differences in personality. In this study, “inhibitory risk” refers to the risks that make it difficult to continue healthy living habits due to a sense of oppression stemming from one’s living environment and lifestyle.

Many studies examining subjects’ maintenance of healthy living habits have assessed positive perspectives to explore methods for promoting them [20,21]. Maintaining healthy living habits requires not only their promotion but also the suppression of inhibitory risks. This perspective is important in considering why people do not implement living habits as they would rationally [22]. The results of existing research show that factors of inhibitory risks, such as ‘economic constraints’ influence the sustainability of walking habits [23]. Other studies have concluded that ‘living environment’ influences nutritional balance habits, such as the excessive consumption of fast food [24,25]. However, no report in the relevant literature describes a study of differences in the risk factors from the perspectives of both demographics and personality traits.

### 1.2. Purpose

The purpose of this study is to research the differences in the risk factors of living habits based on personality traits for groups with demographic factors, including age and gender (hereafter, ‘groups’). The ultimate purpose of this study is to explore risk factors from multiple perspectives of ‘subjective perception’, ‘individual lifestyle’, to ‘social and community network’ and ‘living environment’. An additional purpose is to propose a new health promotion policy supporting the maintenance of healthy lifestyle habits.

### 1.3. Research Features

Characteristics of this study are the following:1)This study is novel in that it analyzes the difference between the risk factors of living habits based on both demographic factors and personality traits.2)This study assesses risk factors and provides reference information for future planning of healthy cities based on a comprehensive perspective incorporating subjective perception and the living environment.3)This study proposes solutions to the difficulties of improving living habits and their ‘reversion,’ by researching online questionnaires.

## 2. Materials and Methods

### 2.1. Survey Method

It is necessary to ascertain (1) demographic factors and personality traits, and (2) risk factors of residents. For this study, the questionnaire was designed referencing current healthy living habits in Japan, and it was created referencing 146 items in four major constructs: demographics, personality characteristics, and risk factors. Table A1 shows the general survey contents. 

This research commissioned the Rakuten Insight Company to distribute the questionnaire online. The reason for choosing Rakuten Insight Company in this study is that they have approximately 2.3 million qualified market research respondents in Japan [26]. In addition, the author had commissioned the company to distribute another questionnaire online in 2016 and obtained highly reliable data [27]. For the above reasons, this study used Rakuten Insight Company in addition to conducting a random sample survey of its registered monitors. The research was designed to ensure that the survey data for lifestyle habits was representative of daily life, therefore the questionnaire was distributed during non-long-term holidays. The specific dates were in the fall of 2018, September 21st through 28th.

#### 2.1.1. Participants & Procedure

For the participants, a total of 20,000 samples were collected. The effects of transportation conditions such as vehicle ownership were considered with regard to region size, and as such, locations of various sizes were selected. Tokyo (a megacity-city with a well-established public transportation system) and six Kanto counties (provinces and cities around Tokyo) were selected for the study. They are areas of various scale, ranging from dense urban central city areas to surrounding suburban city areas and villages. This eliminated the skewing of the online questionnaire brought by the city scale, including effects on age and gender. Specifically, the results for the three kinds inhabited city scales, three age groups, and two genders were divided specifically into 18 cells, with each cell having the same number of samples. Moreover, for the statistical analysis of each stage of behavioral change addressing each living habit, the established system collected the same quantity for each unit.

According to the description above, this study extracted 1000 target samples from 20,000 samples. Furthermore, the invalid samples (incomplete questionnaires and those with invalid answers (all answers select the same option, for example, all options select the A.option.)) were disposed of, resulting in 954 valid target samples that were retained for analysis. 

#### 2.1.2. Measures

All questions in the survey were close-ended. Participants responded to personality trait items using a 7-point Likert Scale (from 1 = “Strongly disagree” to 7 = “Strongly agree”) [28,29]. In the Definitions section, the author introduces the specific composition and basis of the questionnaire.

### 2.2. Definitions

#### 2.2.1. Definition of Living Habits

To explore the risk factors to health, it was necessary to select representative living habits for analysis. This study, based on evaluation of improved lifestyle habits in the mid-term report of ‘Kenkou-Nippon 21’, used a questionnaire to elicit responses from which we extracted living habits that had not improved for several years or had reverted. This study analyzes and investigates ‘nutritional balance’, ‘sleep’, ‘exercise’ and ‘walking’ as living habits. The specific definitions respective these are as follows:(nutritional balance): two or more meals per day, including staple foods, a main course, and side dishes.(In the first question of this survey, definitions and some examples of staple foods, a main course, and side dishes were provided as a reference to the survey respondents.)(sleeping): adequate sleeping time (longer than seven hours).(exercise): exercise regularly (over 30 min each time).(walking): walk 8000 steps or more each day.

In the living habits items, this questionnaire used objective question (e.g., “Do you walk 8000 steps or more each day? Please choose one from the following opptions which are true.”).

#### 2.2.2. Definition of Stages of Behavioral Change

This study adds a ‘reversion’ stage to the five stages of behavioral change of lifestyle habits proposed in DiClemente’s research [10]: ‘pre-contemplation’, ‘preparation’, ‘contemplation’, ‘action’ and ‘maintenance’. Before the formal survey, a pilot study was conducted to ascertain whether the sample for stages of behavioral change of living habits could satisfy the conditions for statistical analysis. Those survey results show that the number of respondents for ‘contemplation’ and ‘preparation’ was too low to satisfy this condition. Therefore, for the actual study, ‘contemplation’ and ‘preparation’ were combined as ‘preparation’ to establish the five stages of behavioral change in this study. The specific definitions of the stages of behavioral change are shown below:(pre-contemplation) stage: no desire to improve, no action plan.(preparation) stage: want to improve and having a plan, but no action.(action) stage: action, lasting more than 1 month.(maintenance) stage: sustaining action indefinitely.(reversion) stage: was in action for more than a month, but stopped.In the items referencing the stages of behavioral change, this questionnaire used objective question. (e.g., “In the last three years, which of the following descriptions did your walking habits follow? Please choose one from the following opptions which are true.”).

#### 2.2.3. Definition and Constitution of Attributes

In this study, multiple attributes included not only demographic factors such as age and gender but also characteristics of personality traits. The structure of demographic factors and personality traits were as follows: (Demographic Factors):1)Gender,2)Age group (young residents, 25–44 years old; middle-aged residents, 45–64 years old; elderly residents, over 65 years old [6],3)Household type (single household, couple household, multi-generational household),4)Employment status (employed, unemployed),5)State of vehicle ownership (no ownership or cannot use any vehicle, hereafter, ‘no vehicle’), self-specific owned vehicle (hereafter, ‘self-ownership’), non-self-owned vehicle, but an available family car or car-sharing service (hereafter, ‘co-ownership’), and6)Living city scale (central city—Tokyo 23 wards, Chiba city, Yokohama city, Kawasaki city, Sagamihara city, Saitama city); cities in central city commuting area [30], Medium-sized cities (hereinafter, ‘surrounding cities’); small cities and villages (other than the central cities and surrounding cities).

In the stages of demographic factors items, this questionnaire used objective question (e.g., “Please choose your gender (Male/Female)”).(Personality Traits):To measure personality traits related to health status, the study used the hardiness scale [26]. The hardiness scale measures three personality trait dimensions: challenge, control, and commitment. Challenge refers to the tendency to view unexpected changes or potential threats as positive challenges for future growth, rather than as harmful events. Control refers to the awareness that control over daily life events is possible. Commitment’ refers to the tendency to feel committed and involved in different areas of life. The Japanese version of the scale has been verified as reliable [27]. In the stages of personality traits items, this questionnaire used a 7-point Likert Scale (e.g., “Q. Most days, life is really interesting and exciting for me. “1 = “Strongly disagree” to 7 = “Strongly agree””).

#### 2.2.4. Risk Factors

Before implementing the survey, it was necessary to presuppose the risk factors from multiple perspectives. Therefore, this study refers to the three hierarchies of the ‘social determinants of health model’, such as ‘living environment’, ‘social and community network’ and ‘individual lifestyle’ developed for the public health field [31]. Furthermore, based on the results of earlier studies, the subjective perception of individuals is an important factor influencing living habits [32]. Therefore, this study adopts the ‘subjective perception’ of living habits as a factor of inhibitory risk. Moreover, combined with the items above, the survey questionnaire about risk factors is designed from four items. Moreover, ‘constraints of individual lifestyle’ are divided into ‘time constraints’ and ‘economic constraints’. Also, ‘subjective perception’ is divided into ‘sense of health’ and ‘negative awareness’. Table A1 presents the question contents for the respective items. In the risk factors items, this questionnaire used objective question (e.g., “Why are you not maintaining your walking habits? Please choose all of the following opptions which apply to you.“).

### 2.3. Analytical Approach

Based on data from the questionnaire survey, this study proposes a policy to promote healthy city development according to the results of the following three-step analysis:1)Ascertaining personality traits through factor analysis: To extract the factors of personality traits from the 15-item hardiness scale. Referring to the methods of research [26], this study adopted the factor analysis method. The reason for not using principal component analysis is that there is no affiliation between the questions in the 15-item hardiness scale, therefore it does not apply to principal component analysis.2)On the basis of Analysis step-1, the crosstabs analysis was used, exploring changes in the distribution tendency of stages of behavioral change of various living habits with different demographic factors and personality traits; also, analyzing correlation between demographic factors and personality traits of the ‘non-action’ or ‘maintenance’ stages of behavioral change, such as ‘reversion’, ‘preparation’, and ‘pre-contemplation’ stages.3)Investigation as to whether there is a correlation between the individual attributes and the sensitivity of the risk factors of each lifestyle was done. Extracting and grouping demographic factors that affect inhibitory risk (divided into 21 groups and explained in the following sections). It was done based on the above grouping results in each group, with the ratio of personality traits as X, incidence frequency of the risk factor as Y, and a single regression analysis being performed to analyze the effects of differences in personality traits of each group on the maintenance or action of each living habit.

## 3. Results

### 3.1. Results of the Questionnaire Survey

The collection status is shown in Figure 1. The sample size of each gender, age group, and city size is controlled when the target samples is taken. According to the recycling survey results, the sample size of males (=501) and females (=453) is almost the same. However, due to the substantial difference in the number of registered monitors in different age groups and city sizes, this study controlled the deviation of the number of samples recovered in each age group (younger residents = 408, middle-age residents = 378, elderly residents = 168) and inhabited city scale (central cities = 400, surrounding cities = 388, small-sized cities and villages = 166)to the maximum extent. The distribution of other non-control, demographic factors is mostly in the sample of employed (=677). and multi-generational-household (=498). In the case of the state of vehicle ownership, there is little difference in the sample of no vehicle (=319), self-ownership (=396), and co-ownership (=239). Samples of each stage of behavioral change living habits were also controlled, although maintenance stages of the nutrition balance and sleeping contained more samples than the other samples. 

### 3.2. Results of Personality Attribute Analysis

To ascertain characteristics of the internal attributes, and to verify the survey, data of personality traits is statistically reliable in this study. Factor analysis was applied to the 15 items on the left side of Table 1. Based on the cumulative rate and load amount, three personality traits challenge, control, and commitment were extracted similarly to the earlier studies. In previous studies, the two items, ‘I like to learn new things’ and ‘I like to change my daily schedule’ are classified as control features. However, for the present study, the two items are classified as challenge features according to the cumulative rate. In other words, the findings of the personality traits of this study are trustworthy and can be referred to past research, and the extracted personality traits are classified into the following three factors, Challenge, Control, and Commitment.

### 3.3. Relation of Stage Change, Personality Traits, and Demographic Factors

Next, a cross-tabulation analysis was conducted with the distributional trends of the stages of behavioral change for each living habit because due to different demographic factors and personality traits. Results of demographic factors are presented in Figure 2. Results of the personality traits are depicted in Figure 3. To ease comparison of the results of each cross-tabulation, a percentage of the data bars is added to each grid. The percentage refers to the proportion of people at each stage of behavioral change in the total number of people in a kind of demographic factors or personality traits.1)Specifically examining the results of the vehicle ownership status in Figure 2, the living habits of those who use vehicle co-ownership do not tend to stay in ‘pre-contemplation’ stage. They are more inclined to be in the ‘action’ stage of improving living habits.2)Residents of single households are less inclined to adopt healthy lifestyles.3)Keep their living habits, but residents with multi-generational-households tend to ‘maintain’ their stages of nutritional balance, sleep, and exercise.4)Based on the above results, residents who use co-vehicle are more likely to have partners who work together. This kind of residents is more inclined to improve their living habits because of the help of their peers. Contrary to the single households, it is difficult to recognize that they need to improve current living habits and lack of support for living habits due to the lack of influence of another person.5)Young residents tend not to ‘maintain’ nutritional balance, sleep, and exercise habits. They tend to stay in the ‘preparation’ stage.6)Employed residents are more likely to be in the ‘reversion’ stage of nutritional balance than unemployed residents.7)Young residents and employed residents are more inclined to be more busy working, therefore it is more difficult for them to maintain healthy living habits in the long-term. In addition, although it is likely that they appear as though they are trying to change living habits, however it is more likely that they are in the ‘reversion’ stage.8)Specifically examining the results of the personality traits in Figure 3, each living habit of those with a strong challenge feature and commitment feature tend to be in the ‘maintain’ stage or the ‘action’ stage. People with a strong ‘control’ feature have experienced less of the ‘reversion’ stage of nutritional balance and exercise.9)According to the above analysis of personality traits, residents with challenge features are more inclined to try new things, and those residents with commitment features are more inclined to have a more positive attitude toward the change of living habits. Therefore, it can be considered that residents with challenges and commitments features are prone to improve behavioral change and more positive attitudes towards improvement, so that improved stage of behavioral change to the ‘action’ stage, and keeping in ‘maintenance’ stage. Residents with the ‘control’ feature are more likely to believe that they will get the desired results according to their efforts. Therefore, such residents are less prone to the ‘reversion’ stage.

Subsequently, the study analyzed correlations in the sensitivity of inhibitory risk with demographic factors and personality traits for residents who did not take actions to improve their living habits, being in the ‘reversion’, ‘preparation’, or ‘pre-contemplation’ stage. This study assesses whether the sensitivity to inhibitory risk depends on whether the number of choices in Table A1 is greater than the average number of choices, or not. The analysis results are presented in Table 2.1)Non-elderly residents who are in ‘reversion’, ‘preparation’, and ‘pre-contemplation’ stages tend to feel inhibitory risk.2)Specific analysis of the state of vehicle ownership revealed that residents who use vehicle co-ownership easily feel inhibitory risks of sleep and exercise living habits.3)For the analysis of personality traits, stronger the challenge and commitment features are associated with greater likelihood of sensitivity to inhibitory risk. However, residents with strong control features tend not to feel inhibitory risk. Residents with strong control features devote more attention to achieving their goals through their own efforts. Therefore, they tend to report that they might not be readily aware of external inhibitory risk.

### 3.4. Relations of Risk Factors and Personality Traits by Extrinsic Personal Attribute Group

Based on results represented in the previous section, we find that residents who are in the ‘reversion’, ‘preparation’ and ‘pre-contemplation’ stages of each living habit have different sensitivity to inhibitory risks because of differences in demographic factors and personality traits. In this section, according to the analysis method described in (3) of Section 2.3, the factors of demographic factors that are sensitive to inhibitory risks are grouped. Therefore, the respondents are classified in terms of ‘reversion’, ‘preparation’, and ‘pre-contemplation’ stages of each living habit, as well as state of vehicle ownership and age groups of demographic factors. Furthermore, because of the small number of respondents, the elderly residents are classified irrespective of the vehicle ownership status. Group results and group numbers are presented in Table 3.

To calculate the sensitivity of the constituent factors of inhibitory risks in each group and the strength of personality traits, this study uses the following two equations.1)Equation (1) measures the incidence of the factors of inhibitory risks of living habits (items presented in Table 1) in each group. The higher the incidence frequency, the more likely a person is to feel the inhibitory risks related to corresponding living habits:(1)$$R_{CFgij}=\frac{CF_{gij}}{CF_{gj}}$$$R_{CFgij}$: incidence of i risk factors of g group of j living habit$CF_{gij}$: number of i risk factors of j living habit of g group$CF_{gj}$: number of j living habit of g groupg: 1–21 groupsj: nutritional balance (N), sleeping (SL), exercise (EX), walking (W)i: living environment (EM), social and community network (SN), individual lifestyle (time (T) · economic (EC)), subjective perception (sense of health (H) · negative awareness (N))2)Equation (2) is used to measure the ratio of each feature of personality traits. Higher ratios are associated with stronger personality traits of the group:(2)$$R_{Pgk}=\frac{P_{gk}}{CF_{gj}}$$$R_{\mathrm{Pgk}}$: ratio of g group’s stronger k feature of personality traits$P_{\mathrm{gk}}$: number of g group’s stronger k feature of personality traitsk: challenge (CHA), control (CON), commitment (COM)

Furthermore, for each group, simple linear regression is used to analyze whether correlation exists between the incidence of factors of inhibition risk $R_{CFgij}$ and the ratio of personality traits $R_{Pgk}$. The results are presented in Table 4. 1)Significant correlation was found between the ratio of challenge feature and the incidence of factors of inhibitory risk.2)The risk factors for walking habits are more likely to consist of multiple items.3)Compared to other personality traits, residents with challenge features are more likely to try new things. Therefore, such residents need to think about which aspect they can change. At the same time, they are more likely to be exposed to aspects that they cannot change in the process of thinking, which is the risk factor in this study. For example, in order to change their exercise habits, residents showing the challenge feature will think about their own schedules and expenditures, and adjust their schedules to set aside exercise time or save money for sports. Among them, some residents may realize that they can’t make adjustments in their work schedule and realize that time is an inhibitory risk.4)Residents with a commitment feature do not have an inclination to challenge new things, therefore making it difficult for them to realize what obstacles in their living habits prevent them from maintaining a healthy lifestyle. Residents with control feature are more inclined to rely on their own efforts, rather than being easily influenced by external factors.

As can be seen in the analysis results in Table 4, this analysis created scatter plots to examine and compare the correlation between the groups. 21 groups classified with age group, state of vehicle ownership, and stage of behavioral change, with the number of these groups being shown in Table 3. Also, personality traits and risk factors were addressed (for example, risk factors for living environment, risk factors for social networks, etc. The detail of risk factors are organized in Table A1). 

In addition, each living habits corresponds to its scatter plot results. Wherein, the horizontal axis is the ratio of group’s personality traits, and the vertical axis is the incidence of risk factors of group of living habit. According to the analysis results of Table 4, only the challenging feature which are statistically correlated with the risk factors are selected as the horizontal axis in the scattergram. And only the ratio of the risk factors that are relevant to the challenging feature is extracted as the vertical axis. 

For a clearer comparison, the results of each living habit is separated by age, and the points with different ratios of different risk factors in each age group are displayed in the scatter plot. The final extracted result is presented in Figure 4 (underlined in Table 4).

Overall, irrespective of the group, the higher the ratio of the challenge feature, the higher the incidence of risk factors in each living habit. In contrast to residents in the ‘pre-contemplation’ stage, residents in the ‘reversion’ stage tend to have a higher ratio of the challenge feature and are more sensitive to a higher incidence of risk factors (Figure 4).1)Specifically examining the results of nutritional balance (Figure 4a,b), among those in the ‘preparation’ stage, the incidence of the social and community network between the young residents (10 group) and middle-aged residents (13 groups) who used the co-ownership vehicle shows the largest difference in the inhibitory risk factor (Figure 4a,b).2)Specifically examining the results of sleep (Figure 4c), young residents (g1, g2, and g3 groups) are in the ‘reversion’ stage, and the time constraints of individual lifestyles are higher than that of others. However, the difference in the occurrence incidence of the risk factors of subjective perception in sensory perception might not be readily apparent.3)Specifically examining risk factors in walking habits (Figure 4d1,2), the challenge feature ratio of the elderly residents in the ‘reversion’ and ‘preparation’ stages are lower than that of the non-elderly residents. Moreover, the incidence of risk factors is low. However, elderly residents in the ‘preparation’ stage have a high incidence of negative awareness (Figure 4d1). No significant difference was found in the ratio of different challenge features because of stages of behavioral change in elderly residents.

## 4. Discussion

Based on the results of the analysis, the following proposals can be made:1)Specifically addressing the analysis of vehicle ownership status, residents who use vehicle co-ownership tend not only to remain in the ‘pre-contemplation’ stage but also tend to be in the ‘action’ stage of improving each living habit. Moreover, couple households can often maintain nutritional balance, sleep, and exercise habits. Therefore, the presence of other people, such as acquaintances and partners, might be an important factor to improve living habits. Previous research has shown that social networks have an impact on improving health issues such as obesity [1]. In that study, the social network did not directly affect the improvement of diet and exercise behavior but was affected by surrounding friends or family members. Colagiuri’s research results corroborated the results from this study. Whether Australia or Japan, it is not limited to countries or regions that social networks have a positive impact on maintaining or improving living habits. Furthermore, policies prescribing the promotion of social communities that can provide partners for single-person households could result in revolutionary changes. In combination with the results of vehicle ownership status, promoting the car sharing service within the community or within a certain range can promote the formation of the surrounding social networks, and can eliminate obstacles caused by the inconvenience of public transportation.2)Residents who use vehicle co-ownership tend to be sensitive about the risk factors in their sleep and exercise habits. The reason might be due to limitations to their behavior and time due to anxiety associated with traffic. Therefore, improving the public transportation service level and thus encouraging free movement could be more effective in improving sleep and exercise habits.3)Comparison with unemployed residents shows that employees tend to a ‘reversion’ stage in nutritional balance living habits. Establishing a balanced diet in a company is an important policy. In other research, it has been proved that providing healthy foods such as fruits in the workplace is more likely to interfere with negative nutritional living habits of the members, thereby improving the effect of chronic diseases [33].4)Compared with the demographic factors, the correlation between the personality traits and risk factors in each living habit are might be readily apparent. Policies might exist that are based only on demographic factors such as vulnerable road users, which might result in weaker living habit improvement measures. Therefore, to improve the improvement measure effectiveness, one must implement policies that foster challenging features of residents and formulate new policies based on combinations of challenge features. According to an earlier study [33], during 15 weeks of testing, the tester realized that the need to achieve the set small improvement goals can help to cultivate challenge features of personality traits. Results of this study demonstrate that, for activities of health mobility management or school learning activities, it is easier to improve living habits by combining a policy for individual improvement and the introduction of incremental improvement goals.5)Young residents are stronger in terms of challenging characteristics than middle-aged residents. They are susceptible to peer pressure in the presence or absence of people who eat together and the behavior of surrounding people. Therefore, even if they want to improve their nutritional balance, such behavior might not occur. Action improvement of young residents might increase by offering information not only to individuals but also to the surrounding people. One earlier study [34] found that young people (20s) continue to live dependently on convenience stores and eating in restaurants. They should be educated about ‘proper dietary health’ such as eating vegetables while using convenience stores and eating instant noodles. With that proposal, effects can be improved further if people are educated in ‘proper dietary health’, and if they are targeted not only as one individual among the young residents but as belonging to an entire family or social network that should be using ‘proper dietary health’. There are also other studies on How to used personality traits opinion to improve living habits such as diet and exercise [35,36].6)Elderly residents over the age of 65 are mostly in retirement. Therefore, for elderly residents, the risk factors from time constraints and economic constraints are less binding. However, this study showed that elderly residents in the ‘preparation’ stage often suppress maintenance or improvement of their living habits because of their negative feelings about living habits. For instance, elderly residents are more inclined than non-elderly residents to feel difficulty related to walking and do not feel the necessity to walk. Healthy city policy formulated for elderly residents who are in the ‘preparation’ stage should specifically examine how to smoothly convert negative feelings regarding walking habits, and to convey the importance of walking for health. Thus, the effects of implementing such healthy city policies might increase in effectiveness.

## 5. Conclusions

This study clarified factors of personality traits and found differences in the risk factors of maintenance and improvement of living habits from the viewpoint of multiple attributes combining the demographic factors with personality traits. Results clarified that the correlation between personality traits and stages of behavioral change and the risk factors to living habits are stronger than demographic factors. Furthermore, due to the ‘reversion’ stage of behavior, results show that those in this stage have stronger challenge features than those belonging to other stages of behavioral change. They are more likely to feel inhibitory risk due to demographic factors.(e.g., Based on the results in Table 2, focusing on correlation between the risk factors and demographic factors and personality traits. The value of significance(P) about the three personality traits is less than 0.01, higher than the correlation of demographic factors). It is necessary to not only address demographic factors but to understand that personality traits can suppress the ‘reversion’ stage. With this knowledge, measures for planning healthy cities that can implement and promote sustainable healthy living habits can effectively be made.

Because we have not yet investigated behavioral changes of respondents after new policies are introduced, new policy proposals must be made after investigating respondents’ satisfaction rates with the possibility of participation in future policies promoting healthy lifestyles.

## Figures and Tables

**Figure 1 ijerph-16-02322-f001:**
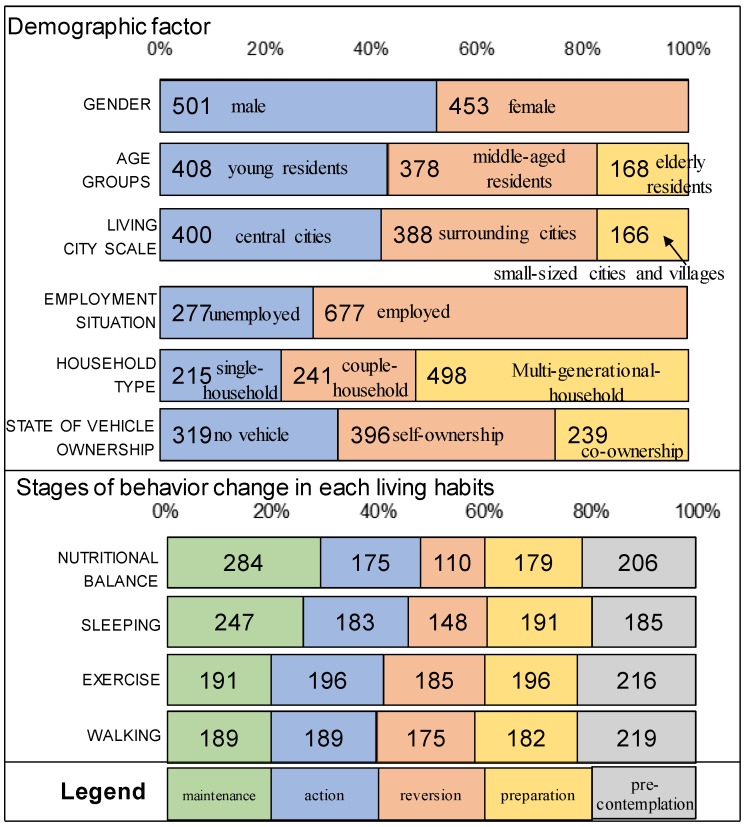
Number of samples of each demographic factor and each stage of behavioral change (*N* = 954).

**Figure 2 ijerph-16-02322-f002:**
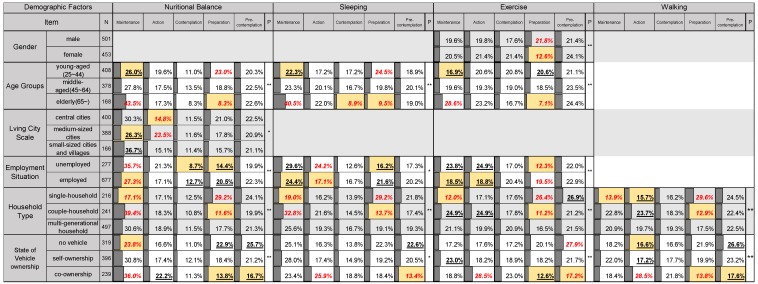
Result of cross-tabulation analysis of the distributional trends of stages of behavioral change of each living habits because of different demographic factors (independence test, *p* value < 0.05 *, *p* value < 0.01 **; only the results which have significant difference are shown. Red numbers: *p* value < 0.01 in residual analysis, **Underlines:**
*p* value < 0.05 in residual analysis. Yellow frame: significantly less).

**Figure 3 ijerph-16-02322-f003:**
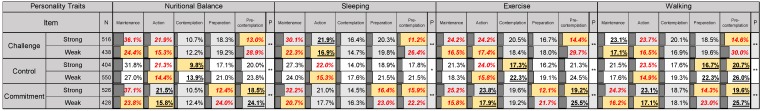
Result of cross-tabulation analysis of the distributional trends of stages of behavioral change of each living habit because of different personality traits (independence test, *p* value < 0.05 *, *p* value < 0.01 **; Red numbers: *p* value < 0.01 in residual analysis, **Underlines:**
*p* value < 0.05 in residual analysis. Yellow frame: significantly less).

**Figure 4 ijerph-16-02322-f004:**
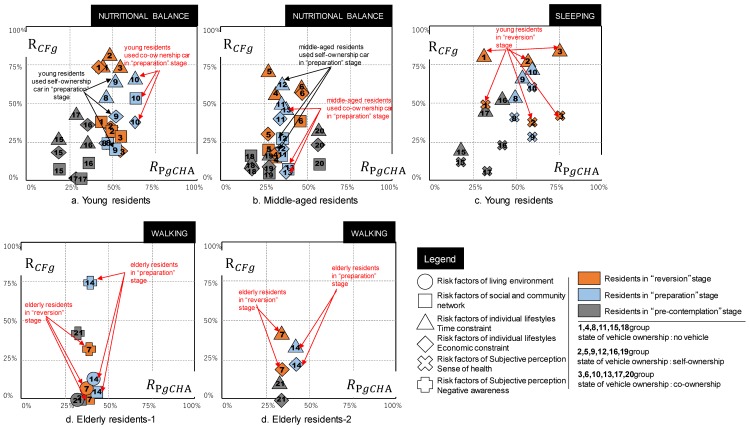
Personality traits (challenge feature) by group and distribution of inhibition risk factors for each lifestyle habit.

**Table 1 ijerph-16-02322-t001:** List of 15-item hardiness scale and factors of personality traits extracted.

No.	15-Item Hardiness Scale	Factors of Personality Traits ^1^		
		Challenge	Control	Commitment
1	Most days, life is really interesting and exciting for me.	0.820	0.182	0.056
2	I find it challenging to do more than one thing at a time.	0.816	0.251	0.051
3	Changes in daily activities are interesting.	0.810	0.307	0.034
4	I would like to have various experiences.	0.808	0.308	0.055
5	I like to change my daily schedule.	0.696	0.313	0.022
6	I like to learn new things.	0.667	0.177	0.170
7	By working hard you can nearly always achieve your goals.	0.226	0.856	0.090
8	How things go in my life depends on my own actions.	0.241	0.850	0.086
9	My choices make a real difference in how things turn out in the end.	0.268	0.816	0.143
10	When dealing with some difficult task, I know when to ask for help.	0.205	0.785	0.045
11	Planning ahead helps to avoid most future difficulties.	0.297	0.719	0.171
12	Something of interest might happen.	0.573	0.048	0.453
13	I don’t think there’s much I can do to influence my own future.	−0.111	−0.052	−0.851
14	I feel that my life is somewhat empty of meaning.	0.087	−0.096	−0.831
15	Most of my life is spent doing meaningful things.	0.343	0.310	0.596

^1^ Factor extraction method: principal component analysis; cumulative ratio: 68.10%.

**Table 2 ijerph-16-02322-t002:** Distribution of susceptibility risk of susceptibility risk and test results of independence.

Category			Nutritional Balance			Sleeping			Exercise			Walking		
**Demographic factors**	**item**		*N*	%	*p*	*N*	%	*p*	*N*	%	*p*	*N*	%	*p*
	**state of vehicle ownership**	**no vehicle**				113	60%	* 0.017	82	39%	* 0.019			
		**self-ownership**				140	65%		118	51%				
		**co-ownership**				92	76%		67	53%				
	**gender**	**male**	117	44%		187	66%		139	46%		209	67%	
		**female**	101	44%		158	66%		128	49%		189	72%	
	**age groups**	**young residents**	115	52%	** 0.0001	181	73%	** 0.0001	143	56%	** 0.0001	192	76%	** 0.003
		**middle-aged residents**	88	43%		133	62%		104	45%		151	65%	
		**elderly residents**	15	23%		31	49%		20	25%		55	60%	
	**employment situation**	**unemployed**	40	34%	* 0.012	73	57%	* 0.021						
		**employed**	178	47%		272	69%							
**Personality traits**	**challenge**	**strong**	112	61%	** 0.0001	167	80%	** 0.0001	136	60%	** 0.0001	193	83%	** 0.0001
		**weak**	106	34%		178	57%		131	38%		205	60%	
	**control**	**strong**	95	37%	** 0.001	161	58%	** 0.0001	126	42%	** 0.010	180	59%	** 0.0001
		**weak**	123	52%		184	75%		141	53%		218	80%	
	**commitment**	**strong**	94	53%	** 0.003	146	73%	** 0.009	118	54%	** 0.010	174	77%	** 0.0001
		**weak**	124	39%		199	61%		149	43%		224	64%	
**Average check count of constituent factors of inhibition risk**			2.08			1.7			2.25			1.8		
**Checked number of constituent factors of inhibition risk**		**large**	277			345			300			398		
		**less**	218			179			267			178		
**Total (n)**			495			524			567			576		

^1^ By demographic factors and personality traits, (*N* *%) shows only the results in which the checked number of inhibition risk factors is greater than average; % = the samples of checked numbers is more than average in each attribute/sample of each attribute. Independence test, *p* value < 0.05 *, *p* value < 0.01 **. Only the results which have significant difference are shown.

**Table 3 ijerph-16-02322-t003:** Sample number (*N*) and group number.

Classified Items			Nutritional Balance	Sleeping	Exercise	Walking	Group No.
**「reversion」stage**	**young residents**	**no vehicle**	11	22	21	24	1
		**self-ownership**	23	27	38	33	2
		**co-ownership**	11	21	26	19	3
	**middle-aged residents**	**no vehicle**	18	16	22	18	4
		**self-ownership**	20	26	27	28	5
		**co-ownership**	13	21	23	26	6
	**elderly residents**		14	15	28	27	7
**「preparation」stage**	**young residents**	**no vehicle**	41	40	35	36	8
		**self-ownership**	34	36	32	35	9
		**co-ownership**	19	24	17	20	10
	**middle-aged residents**	**no vehicle**	23	23	23	23	11
		**self-ownership**	35	34	36	37	12
		**co-ownership**	13	18	11	11	13
	**elderly residents**		14	16	12	20	14
**「pre-contemplation」stage**	**young residents**	**no vehicle**	40	37	44	41	15
		**self-ownership**	26	25	24	27	16
		**co-ownership**	17	15	18	17	17
	**middle-aged residents**	**no vehicle**	26	22	27	26	18
		**self-ownership**	43	42	46	47	19
		**co-ownership**	16	12	16	16	20
	**elderly residents**		38	32	41	45	21

**Table 4 ijerph-16-02322-t004:** Correlation between the incidence of factors of inhibition risk $R_{CFgij}$ and the ratio of personality traits $R_{\mathrm{Pgk}}$ in each group.

Living Habits	Explanatory Variable				Objective Variable ^1^					
	Category		ALL		Challenge		Control		Commitment	
		Subcategory	*N*	%	*R* ^2^	*p*	*R* ^2^	*p*	*R* ^2^	*p*
**Nutritional balance**	**Constraints in living environment**		80	16%	0.27	0.02	0.00	0.84	0.03	0.43
	**Constraints due to social and community network**		88	18%	***0.41***	0.00	0.01	0.75	0.00	0.81
	**Personal lifestyle constraints **	**Time constraints**	219	44%	***0.42***	0.00	0.04	0.39	0.01	0.75
		**Economic constraints**	114	23%	***0.30***	0.01	0.04	0.41	0.07	0.24
	**Subjective perception**	**Sense of health**	71	14%	0.13	0.11	0.14	0.10	0.02	0.54
		**Negative feelings**	228	46%	0.27	0.01	0.06	0.28	0.02	0.51
**Sleeping**	**Constraints in living environment**		58	11%	0.02	0.55	0.04	0.41	0.00	0.90
	**Constraints due to social and community network**		-	-	-	-	-	-	-	-
	**Personal lifestyle constraints **	**Time constraints**	265	51%	***0.46***	0.00	0.00	0.86	0.02	0.57
		**Economic constraints**	87	17%	0.23	0.03	0.00	0.97	0.01	0.63
	**Subjective perception**	**Sense of health**	148	28%	***0.52***	0.00	0.00	0.92	0.02	0.59
		**Negative feelings**	185	35%	0.27	0.02	0.00	0.96	0.04	0.36
**Exercise**	**Constraints in living environment**		104	18%	0.24	0.02	0.03	0.47	0.08	0.22
	**Constraints due to social and community network**		50	9%	0.21	0.04	0.01	0.62	0.05	0.34
	**Personal lifestyle constraints **	**Time constraints**	273	48%	***0.42***	0.00	0.02	0.51	0.05	0.33
		**Economic constraints**	131	23%	0.28	0.01	0.03	0.45	0.00	0.95
	**Subjective perception**	**Sense of health**	100	18%	0.26	0.02	0.04	0.39	0.04	0.39
		**Negative feelings**	281	50%	0.26	0.02	0.02	0.51	0.06	0.29
**Walking**	**Constraints in living environment**		89	15%	***0.30***	0.01	0.07	0.26	0.03	0.43
	**Constraints due to social and community network**		41	7%	***0.36***	0.00	0.00	0.84	0.00	0.94
	**Personal lifestyle constraints **	**Time constraints**	254	44%	***0.49***	0.00	0.01	0.65	0.03	0.49
		**Economic constraints**	85	15%	***0.32***	0.01	0.03	0.46	0.03	0.42
	**Subjective perception**	**Sense of health**	99	17%	0.19	0.05	0.04	0.42	0.02	0.53
		**Negative feelings**	280	49%	***0.40***	0.00	0.00	0.87	0.18	0.06

^1^ Result of the linear regression with response variable, $R_{CFgij}$; explanatory variables, $R_{Pgk}$; *R^2^*, coefficient of determination; *R^2^* ≥ 0.3, **Underlines**; *p* value: significance.

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
