# Peer review of "Inhibitory Risks Affecting the Maintenance of Healthy Lifestyle Habits—A Study Based on Demographic Factors and Personality Traits"

_ijerph, 2019, doi:10.3390/ijerph16132322_

Round 1
Reviewer 1 Report
The manuscript has promise. While the paper could offer useful insights for healthcare professionals, urban planners and/or policy makers, it is still not “there”. To help the reader better understand and contextualize the findings, the manuscript will require substantially more work. I have provided detailed feedback below that I believe may improve the paper. .
Basic Issues
Grammar/Formatting: The authors may want to check the spacing of the words and special characters. Example: Line 23 “time.(‘Challenge’)” etc.
Terminology: I suggest the authors use “risk factors” or “social risk factors” instead of “inhibitory risk” or “inhibitory risk factors”. The word “inhibitory” is somewhat confusing.
Suggestions for Specific Manuscript Sections
Methods
The method section is somewhat confusing. I would suggest reorganizing this section and following standard manuscript conventions. Maybe create four subsections: Participants/Measures/Procedures/Analytical Approach (or something along those lines).
Participants: à give some background verbatim here using the information from Figure 1. A short paragraph should suffice.
Measures: Please describe the questionnaire in more detail and how it was constructed (e.g. how many total items, what constructs were being measured etc,) To do that you could simply rewrite your definitions (lines 114-179) and condense the information from tables 1 and table 2 here. I would also delete tables 1 and 2 because they really do not add much here in the methods section. If the authors feel strongly about sharing the actual questionnaire items (and the editor doesn’t mind), I would put them into an appendix or an externally-hosted link.
This could look something like:“A questionnaire was created consisting of 200 items capturing three major constructs: demographics, personality characteristics, and risk factors. …”. à then describe each of the constructs in more details. For example, rewrite manuscript section lines 161-167.
“To measure personality traits related to health status, the study used the hardiness scale [26]. The hardiness scale measures three personality trait dimensions: challenge, control, and commitment. Challenge refers to the tendency to view unexpected changes or potential threats as positive challenges for future growth, rather than as harmful events. Control refers to the awareness that control over daily life events is possible. Commitment’ refers to the tendency to feel committed and involved in different areas of life. The Japanese version of the scale has been verified as reliable [27]. à Then go on to explain the other constructs/questionnaire section …
Procedures: describe what and how you did the study including more specifics on sampling and the survey details. For sampling, consider rewriting lines 182-188. Give your reader ore information on how exactly you sampled. Is it a convenience sample? A purposive/theoretical sample? A random sample? How were the participants approached? Why the authors use this type of sample? What online survey tool was being used etc.? This information will give the reader a better idea about the quality of the sample and a starting point if they wanted to replicate the findings. In addition, I would like to see some more information on how the actual survey was conducted. What was the time frame? Summer 2018? What was the response rate (=percentage of people who took the questionnaire as opposed to the number of people who received the questionnaire). Also, please add any other relevant information (e.g. did participants receive compensation, how long did the survey take, if the questionnaire is not added as link/appendix give a sense of the nature of items: were they all close-ended? Or did the survey contain open-ended questions? What Likert scales were used etc., Maybe add any pertinent grant information here as well.)
Analytical Approach: There needs to be much more information how exactly the survey data were analyzed. With some more clarity here, the results section could become less consuming. Here are a couple ideas that might help:
· Discuss why a Principle Component Analysis and not an Exploratory/Confirmatory Factor Analysis was being used and describe the specific purpose of running this type analysis. Give more information about PCA(e.g. discuss whether assumptions were met, what exact procedure was been used (citation would be preferable etc.) à in short: give more info).
· Please also briefly justify the appropriateness of the other analytical approaches. Why, for example, did the authors use crosstabs and “single regressions” rather than some other form of multivariate techniques (e.g. multiple regression). Please also discuss why single regressions rather than correlations were used (line 300). As the authors are probably aware of one is a statistical measure involving predictions (X predicts Y), while the other is a measure of association (X is associated with Y). In addition, please describe what test of independence for the crosstabs was run? A Chi Square Test of Independence? Why not another measure of association like gamma?
Results:
The results section also could benefit from some reorganization and context. At this point, it feels more like a collection of tables/figures and bulletin points. Here are some suggestions for improvement:
· Before discussing the results within subsections, give the reader a BIG PICTURE introduction. Briefly, summarize what the analyses - taken together - seem to demonstrate. Put another way, give the reader a sense of how risk factors seem to be related to the personality traits, demographics and the different stages in behavior change, living habits? What’s the “story” in the data?
· Section 3.1 (line 210ff): It is not clear as to why the author subjected the data to a PCA. What does the factor analysis attempt to accomplish? What does it really show? Are the authors merely trying to see if the factor structure of the hardiness scale holds for this sample? Or are there other reasons? If so, the authors need to explain them.
· Section 3.2 (line 220ff): there is again a lot of information packed into this section and the tables. Maybe again an introductory paragraph that ties the findings a bit together would be helpful here.
o Offer more of an interpretative story (what do the data mean/suggest) for each subsections. Example: Offer more “interpretation: for each subsection 3.2. Letting aside for a moment the fact that a Pearson R or Spearman Correlation Coefficient probably would have been better here, the reader would benefit from having much more context here. What, for example, do the high R2 values between Challenge and Living Habits really mean? Why are there low R2 values for the other personality dimensions? What does that really suggest?
o Figure 4, while novel, again packs a LOT of information. And while the corresponding text tries to make sense of some of the complexity in it (lines 319-334), some overarching narrative to connect the bulletin points would VERY helpful for the reader. The authors may also want to consider running a cluster analysis as it may be more robust statistical technique to explore these patterns. Just a thought; not a requirement.
Discussion
Much of the discussion section is really not a discussion section but an extension and/or elaboration of the results (what did the authors find – exception: lines 356-361 and 366-367). I would suggest merging the information from the discussion with the results to provide more context the numbers. This would make the results section more readable.
As to the actual discussion section, I would suggest the authors consider creating subsections that explore how their findings compare to existing studies and – in a separate section - discuss any potential policy implications (and a detailed rationale for them). It would probably also be helpful for the authors to discuss whether these recommendations are specific to the cultural context of the study or if they believe their recommendations can be adopted cross-culturally/transnationally. And so why/how?
Author Response
Please confirm the uploaded file that is added modification detail. The following is a simple response to the comments.
Point 1: Grammar/Formatting: The authors may want to check the spacing of the words and special characters. Example: Line 23 “time.(‘Challenge’)” etc.
Terminology: I suggest the authors use “risk factors” or “social risk factors” instead of “inhibitory risk” or “inhibitory risk factors”. The word “inhibitory” is somewhat confusing.
Response 1: Thank you very much for your comment. I have checked and revised the full paper. And I used “risk factors” instead of “inhibitory risk factors”. However, I am not sure if I can modify the title, so the word” inhibitory risk factors” in the title is left behind.
Point 2: Methods:The method section is somewhat confusing. I would suggest reorganizing this section and following standard manuscript conventions. Maybe create four subsections: Participants/Measures/Procedures/Analytical Approach (or something along those lines).
Response 2: Thank you very much for your comment. I revised the method section. Please see the page3-6 line111-241.
Point 3: Participants: à give some background verbatim here using the information from Figure 1. A short paragraph should suffice.
Response 3: Thank you very much for your comment. I added the detail information about participants. Please see the page6 line 246-258.
Point 4: Measures: Please describe the questionnaire in more detail and how it was constructed (e.g. how many total items, what constructs were being measured etc,) To do that you could simply rewrite your definitions (lines 114-179) and condense the information from tables 1 and table 2 here. I would also delete tables 1 and 2 because they really do not add much here in the methods section. If the authors feel strongly about sharing the actual questionnaire items (and the editor doesn’t mind), I would put them into an appendix or an externally-hosted link.
This could look something like: “A questionnaire was created consisting of 200 items capturing three major constructs: demographics, personality characteristics, and risk factors. …”. à then describe each of the constructs in more details. For example, rewrite manuscript section lines 161-167.
“To measure personality traits related to health status, the study used the hardiness scale [26]. The hardiness scale measures three personality trait dimensions: challenge, control, and commitment. Challenge refers to the tendency to view unexpected changes or potential threats as positive challenges for future growth, rather than as harmful events. Control refers to the awareness that control over daily life events is possible. Commitment’ refers to the tendency to feel committed and involved in different areas of life. The Japanese version of the scale has been verified as reliable [27]. à Then go on to explain the other constructs/questionnaire section …
Response 4: Thank you very much for your comment. I added the detail information about the questionnaire. I moved Table 1 and 2 to an appendix. Please see the page3-5 line111-241.
Point 5: Procedures: describe what and how you did the study including more specifics on sampling and the survey details. For sampling, consider rewriting lines 182-188. Give your reader more information on how exactly you sampled. Is it a convenience sample? A purposive/theoretical sample? A random sample? How were the participants approached? Why the authors use this type of sample? What online survey tool was being used etc.? This information will give the reader a better idea about the quality of the sample and a starting point if they wanted to replicate the findings. In addition, I would like to see some more information on how the actual survey was conducted. What was the time frame? Summer 2018? What was the response rate (=percentage of people who took the questionnaire as opposed to the number of people who received the questionnaire)? Also, please add any other relevant information (e.g. did participants receive compensation, how long did the survey take, if the questionnaire is not added as link/appendix give a sense of the nature of items: were they all close-ended? Or did the survey contain open-ended questions? What Likert scales were used etc., Maybe add any pertinent grant information here as well.)
Response 5: Thank you very much for your comment. I added the detail information about the questionnaire including more specifics on sampling and the survey design. Please see the page3-4, line111-145.
Point 6: Analytical Approach: There needs to be much more information how exactly the survey data were analyzed. With some more clarity here, the results section could become less consuming. Here are a couple ideas that might help:
· Discuss why a Principle Component Analysis and not an Exploratory/Confirmatory Factor Analysis was being used and describe the specific purpose of running this type analysis. Give more information about PCA (e.g. discuss whether assumptions were met, what exact procedure was been used (citation would be preferable etc.) à in short: give more info).
· Please also briefly justify the appropriateness of the other analytical approaches. Why, for example, did the authors use crosstabs and “single regressions” rather than some other form of multivariate techniques (e.g. multiple regression). Please also discuss why single regressions rather than correlations were used (line 300). As the authors are probably aware of one is a statistical measure involving predictions (X predicts Y), while the other is a measure of association (X is associated with Y). In addition, please describe what test of independence for the crosstabs was run? A Chi Square Test of Independence? Why not another measure of association like gamma?
Response 6: Thank you very much for your comment. I added the detail information about the analysed method. In addition, I added the reason of using the specific analytical approaches in this paper. Please see the page5-6, line 225-241.
Point 7: Results:
The results section also could benefit from some reorganization and context. At this point, it feels more like a collection of tables/figures and bulletin points. Here are some suggestions for improvement:
Before discussing the results within subsections, give the reader a BIG PICTURE introduction. Briefly, summarize what the analyses - taken together - seem to demonstrate. Put another way, give the reader a sense of how risk factors seem to be related to the personality traits, demographics and the different stages in behaviour change, living habits? What’s the “story” in the data?
Section 3.2 (line 220ff): there is again a lot of information packed into this section and the tables. Maybe again an introductory paragraph that ties the findings a bit together would be helpful here.
Response 7: Thank you for your comment. I reorganized the results section. Please see the page8, line 284-314.
Point 8: Section 3.1 (line 210ff): It is not clear as to why the author subjected the data to a PCA. What does the factor analysis attempt to accomplish? What does it really show? Are the authors merely trying to see if the factor structure of the hardiness scale holds for this sample? Or are there other reasons? If so, the authors need to explain them.
Response 8: Thank you for your comment. I added explanation about the reason why I selected the PCA analysis. In addition, I added detail explanation of the result of the PCA analysis. Please see the page6-7, line 260-268.
Point 9: Offer more of an interpretative story (what do the data mean/suggest) for each subsections. Example: Offer more “interpretation: for each subsection 3.2. Letting aside for a moment the fact that a Pearson R or Spearman Correlation Coefficient probably would have been better here, the reader would benefit from having much more context here. What, for example, do the high R2 values between Challenge and Living Habits really mean? Why are there low R2 values for the other personality dimensions? What does that really suggest?
Response 9 Thank you for your comment. I added the interpretative stories including the interpretation about the high R2 values between Challenge and Living Habits. Please see the page12, line 381-392.
Point 10: Figure 4, while novel, again packs a LOT of information. And while the corresponding text tries to make sense of some of the complexity in it (lines 319-334), some overarching narrative to connect the bulletin points would VERY helpful for the reader. The authors may also want to consider running a cluster analysis as it may be more robust statistical technique to explore these patterns. Just a thought; not a requirement.
Response 10: Thank you very much for your comment. I revised the explanation about Figure 4, including the implication of Figure 4. Please see the page13, line 399-417. In this study, we didn’t use a cluster analysis because classification is not the purpose of this study. Instead, we used a scatter plot ,and it helped to know the correlation among the stages of behavioral change, demographic factors and personality traits.
Point 11: Discussion
Much of the discussion section is really not a discussion section but an extension and/or elaboration of the results (what did the authors find – exception: lines 356-361 and 366-367). I would suggest merging the information from the discussion with the results to provide more context the numbers. This would make the results section more readable.
As to the actual discussion section, I would suggest the authors consider creating subsections that explore how their findings compare to existing studies and – in a separate section - discuss any potential policy implications (and a detailed rationale for them). It would probably also be helpful for the authors to discuss whether these recommendations are specific to the cultural context of the study or if they believe their recommendations can be adopted cross-culturally/transnationally. And so why/how?
Response 11:Thank you for your comment. I reorganized the discussion section. In addition, I made new section to discuss this study and previous studies. Please see the page15-16, line441-451&458-460&483-484&500-503

Reviewer 2 Report
I commend the authors on conducting this important study looking at external and internal factors which can inhibit healthy lifestyles.
The authors could improve the paper by providing examples of terms such as 'challenge' and 'control'. The narrative is sometimes lost in the overuse of technical language. The paper would benefit from a good proofread.
I would also like to see the authors reflect on the strengths and limitations of their work.
Additional comments can be found on the paper PDF.

Author Response
Please confirm the uploaded file that is added modification detail.The following is a simple response to the comments.
Point 1: Change to 'maintain'. I would advise for the paper to be proof read again.
Response 1: Thank you very much for your comment. I changed the word ‘maintenance’ to ‘maintain’.
Point 2: Remove hyphenation
Response 2: Thank you for your comment. I removed hyphenation.
Point 3: Need to elaborate on the questionnaire. Who owns the questionnaire, is it validated and how were respondents recruited to complete these questionnaires?
Response 3: Thank you for your comment. I added information about the questionnaire in the 111th-141th line, page 3 and 142th-145line, page 4.
Point 4: How is nutritional content measured?
Response 4: Thank you for your comment. I added the definition of lifestyle nutritional balance in detail. Please see 154th-156th line, page 4.
Point 5: Define adequate sleep time
Response 5: Thank you for your comment. I added the definition of adequate sleep time in the 157th line, page 4.
Point 6: Can you define what you mean by strong challenge feature.
Response 6: Thank you for your comment. I added the definition of three personality attributes in detail in the 200th -208th line, page 5.
Point 7: Can you define what you mean by strong control features.
Response 7: Thank you for your comment. I added the definition of three personality attributes in detail in the 200th -208th line, page 5.
Point 8: Would benefit from providing confidence intervals for key results.
Response 8: Thank you for your comment. I added the providing confidence intervals inTable 2, page 10.
Point 9: Would be useful to reference wider literature to support discussion. The authors include only 2 references.
Response 9 Thank you for your comment. I added reference papers, and revised the discussion. Please see the 441th -451th line and 458-460, page 15.
Point 10: This conclusion is not clearly explained in the results section.
Response 10: Thank you for your comment. I revised results section and discussion section in line437-451,pp15
Point 11: Good point about improving workplace diet.
Response 11: Thank you for your comment. I added good point about improving work place diet in line458-460, pp.15
Point 12: (Page17 line382-384) Provide examples.
Response 12: Thank you for your comment. I added an example from the result about correlation between the risk factors and demographic factors and personality traits in line501-504, pp.16

Round 2
Reviewer 2 Report
The authors have competently addressed the points made by reviewers. The paper covers an interesting growing area of research which will add to the knowledge base.